# The Purinergic Landscape of Non-Small Cell Lung Cancer

**DOI:** 10.3390/cancers14081926

**Published:** 2022-04-11

**Authors:** Serena Janho dit Hreich, Jonathan Benzaquen, Paul Hofman, Valérie Vouret-Craviari

**Affiliations:** 1Institute of Research on Cancer and Aging (IRCAN, CNRS, INSERM), FHU OncoAge, Université Côte d’Azur, 06108 Nice, France; serena.janho-dit-hreich@etu.univ-cotedazur.fr (S.J.d.H.); benzaquen.j@chu-nice.fr (J.B.); 2CHU Nice, Laboratory of Clinical and Experimental Pathology, Pasteur Hospital, Université Côte d’Azur, 06000 Nice, France; hofman.p@chu-nice.fr; 3Institute of Research on Cancer and Aging (IRCAN, CNRS, INSERM, Team 4), Université Côte d’Azur, 06100 Nice, France; 4CHU Nice, FHU OncoAge, Hospital-Integrated Biobank (BB-0033-00025), Université Côte d’Azur, 06000 Nice, France

**Keywords:** purinergic signaling, ectonucleotidases, antitumor immunity, P2RX7, immunotherapies, lung cancer

## Abstract

**Simple Summary:**

Lung cancer is the most prevalent cancer worldwide with poor overall survival despite many new therapeutic strategies. We discuss in this review the ability of the purinergic landscape to constitute a new potent strategy in the treatment of lung cancer. After defining lung cancer and its current treatments, we present the proteins of the purinergic landscape as well as the mechanisms leading to the production of extracellular ATP, which is at the top of the purinergic signaling chain. We also review the evidence supporting the potency of this strategy through clinical trials dedicated to the proteins of the purinergic landscape.

**Abstract:**

Lung cancer is the most common cancer worldwide. Despite recent therapeutic advances, including targeted therapies and immune checkpoint inhibitors, the disease progresses in almost all advanced lung cancers and in up to 50% of early-stage cancers. The purpose of this review is to discuss whether purinergic checkpoints (CD39, CD73, P2RX7, and ADORs), which shape the immune response in the tumor microenvironment, may represent novel therapeutic targets to combat progression of non-small cell lung cancer by enhancing the antitumor immune response.

## 1. Introduction

Lung cancer (LC) is the most prevalent cancer worldwide, with an estimated 2.1 million new cases and 1.8 million related deaths annually. More than 70% of LC patients are diagnosed at an advanced stage, with a 5-year survival rate of less than 10%, whereas the survival rate for patients with early-stage disease ranges from 50 to 70%. Furthermore, regardless of treatment options depending on the tumor stage (e.g., surgery, chemotherapy, radio therapy, or targeted therapies), the disease progresses in almost all advanced lung cancers and in up to 50% of early-stage cancers. New and effective strategies to improve the outcome of LC include immunotherapies based on immune checkpoint inhibitors (ICIs), which are now used in routine clinical practice.

ICIs have revolutionized the treatment of non-small cell lung cancer (NSCLC). PD-L1 expression is presently the only approved biomarker used in routine diagnostics for stratification of ICI therapies [1]. The ICIs exploit the ability of lung cancer cells to evade recognition by the PD-1 axis by restoring tumor surveillance by the immune system. ICIs approved for antitumor treatment are mainly monoclonal antibodies against PD-1 (nivolumab, pembrolizumab, cemiplimab), PD-L1 (atezolizumab, durvalumab, avelumab), and CTLA-4 (ipilimumab and tremelimumab). These treatments were initially used alone and are now combined with other immunotherapies or chemotherapies [1]. In recent years, this new era of treatments has led to significant improvement in the survival and quality of life of patients with LC. Unfortunately, almost all patients with advanced LC experience progression of the disease regardless of treatment options, underscoring the need for new approaches to treat these patients.

Adenosine triphosphate (ATP) and adenosine are components of the tumor microenvironment (TME). Extracellular ATP (eATP) promotes tumor growth but also immune-mediated tumor eradication, mainly via the well-documented purinergic P2RX7 receptor. Adenosine, on the other hand, is generated from eATP via the ectonucleotidases CD39 and CD73 and is an immunosuppressant that acts at the A2A receptor (A2AR) level [2]. Thus, the purinergic landscape consisting of the P2X and P2Y purine receptors, CD39, CD73, and adenosine receptors (P1 purine receptors) shapes TME immune responses and likely acts “in concert” with immunotherapies. In this review, we focus on LC and discuss recent discoveries about the so-called purinergic landscape and its potential as a therapeutic target for treating patients.

## 2. Source of Extracellular ATP

ATP, often referred to as the main energy source of the cell, is released by mitochondria into the cytoplasm and used by all cellular compartments as fuel. ATP is also present outside the cell and is then referred to as extracellular ATP (eATP). eATP is a damage-associated molecular pattern (DAMP) molecule that is recognized by purinergic receptors and degraded by ectonucleotidases.

The source of eATP is generally associated with various types of injury, including trauma, oxidants, and pathogens [3]. eATP in the tumor microenvironment is present at concentrations of hundreds of micromolar [4]. It is released by activated immune cells and dying cells from the immune system, stroma, and tumor compartments. 

The composition of immune cells of human non-small cell lung cancer (NSCLC) was characterized by flow cytometry. This showed that adenocarcinoma have a higher percentage of CD45+ leukocytes than the distal lung [5]. Thirteen different types of immune cells were identified. They included different subpopulations of T cells, B cells, macrophages, natural killer (NK) cells, dendritic cells (DC), and granulocytes, mainly neutrophils. Each of these cells can release ATP via specific mechanisms, including channel-mediated export, vesicular exocytosis, and cell death (Figure 1).

Pore channels comprise the well described pannexin-1 and connexin-43 proteins assemble to form hexameric membrane structures, but also calcium homeostasis modulator 1 (CALHM1) channels, volume regulated ion channels (VRACs), maxi-anion channels (MACs), and P2RX7 itself. Whereas pore channels allow passive leakage of ATP, vesicular transport is an active mechanism under the control of the vesicular nucleotide transporter (VNUT). VNUT acts in concert with vacuolar proton ATPase (V-ATPase) to accumulate the ATP released from mitochondria within vesicles. The fusion of the vesicles with the membrane is regulated by intracellular Ca^2+^ levels and the soluble N-ethylmaleimide-sensitive factor attachment protein receptor (SNARE) and depends on actin cytoskeleton rearrangement and hypoxia. Regulated cell death represents the 3rd source of eATP which can be caspase-dependent and caspase-independent.

Neutrophils do not represent the most abundant immune cell population, accounting for 8.6% of all CD45+ cells in the TME of NSCLC [5]; nevertheless, their role in eATP production is well documented in the literature. What is the upstream activation of neutrophils? What are the signals and mechanisms that trigger ATP release? In 1998, Colgan’s team showed that activated neutrophils (with fMLP) release 5′-adenosine monophosphate (AMP), which is then converted to adenosine and further promotes endothelial barrier function [6]. At that time, it was hypothesized that the release of adenine nucleotides occurs by direct transport or diffusion across the cell membrane. Today, we know that in addition to pathogenic agents, hypoxia and inflammation are pathophysiological responses that activate ATP release from neutrophils [7,8]. In particular, connexin-43 and pannexin-1 are proteins that form hemichannels involved in the release of ATP. It is worth noting that other anion channels are also involved in ATP export, as shown in Figure 1 and described elsewhere [9,10,11]. A substantial fraction of intracellular ATP is also located in cytoplasmic vesicles, and in addition to pore-forming proteins, vesicular exocytosis represents an independent mechanism responsible for ATP release. This mechanism relies on specialized cytosolic granules that store ATP and fuse with the plasma membrane in response to cell signals to secrete their contents. The vesicular nucleotide transporter (VNUT), encoded by the human SLC17A9 gene, is responsible for the vesicular accumulation of ATP [12]. Once accumulated in the secretory granules, ATP secretion has been shown to depend on intracellular Ca^2+^ levels and SNARE proteins that mediate membrane fusion upon stimulation [13]. The third mechanism responsible for ATP release is based on cell death, which may be random, the result of a biologically uncontrolled process, or regulated, e.g., involving tightly structured signaling cascades and defined molecular effector mechanisms. Under regulated cell death, necroptosis, pyroptosis, NETotic cell death, and lysosome-dependent cell death (which can be associated with autophagic cell death when the lysosome is fused to the autophagosome) trigger rupture of the plasma membrane and subsequent release of intracellular ATP, as described in detail elsewhere [14,15]. 

In addition to neutrophils, macrophages, which make up 4.7% of the immune cell infiltrate [5], can also release ATP. They undergo apoptosis, but also necroptosis, pyroptosis, and parthanatosis, a regulated cell death activated by DNA damage triggered by oxidative stress, eventually leading to necroptosis [16]. Moreover, ATP has been shown to stimulate permeabilization of the lysosomal membrane, which in turn causes the release of cathepsin and subsequent activation of pyroptosis. This cellular mechanism, which depends on the P2RX7 receptor and the Ca^2+^ influx induced by pannexin-1, highlights the existence of a positive feedback loop in macrophages in which eATP promotes the release of ATP [17]. 

T cells represent 47% of CD45+ immune cells and therefore dominate the LC landscape. They are sensitive to eATP that enhances their activation via P2RX4 and P2RX7 stimulation [18]. They also release ATP, being subject to death in damaged tissues and tumors [19]. However, we believe that their chronic activation within the TME likely leads to their exhaustion, and it would not be surprising if this lack of effector function correlates with the prevention of cell death in this particular context. 

In addition to these mechanisms, treatments used to combat tumor growth also promote the release of ATP. For example, conventional chemotherapies used to treat NSCLC (platinum derivatives, paclitaxel, and gemcitabine) or radiotherapies (photochemotherapy and irradiation) increase the release of ATP within the TME [20,21]. Moreover, the ATP, as well as the UTP, released after radiation-induced cell damage and cell death, have been shown to act as pro-metastatic factors in human lung cancer cells [22]. Moreover, human A549 lung cells were reported to release ATP after γ-irradiation, and the released ATP contributed to the DNA damage response (DDR) via the P2RX7, P2RY6, and P2RY12 receptors. Remarkably, radiation sensitization was restored when the activity of these receptors was inhibited, resulting in inhibition of the DDR [23,24]. It is important to note that under chemotherapy and radiotherapy, not only ATP but also many DAMPs are released and induce immunogenic cell death (ICD), which corresponds to cell death that causes an immune response. These DAMPs include calreticulin and heat shock proteins released at the cell surface, the chromatin-binding non-histone protein high-mobility group box 1 and the cytoplasmic protein annexin A1 released from dying cells into the TME, and type I interferons (IFNs) released during de novo synthesis [20]. Once bound to their specific pattern recognition receptors expressed by antigen-presenting cells, they trigger both the innate and adaptive immune systems, resulting in cross-presentation of tumor antigens to CD8+ cytotoxic T lymphocytes, providing critical adjuvant for dying cancer cells [25,26].

Overall, several factors lead to the release of ATP into the extracellular space, resulting in the accumulation of eATP and thus its high concentration in the TME, as mentioned earlier, where its action depends on many factors, including the expression of purinergic receptors and the presence of extracellular ATP-degrading enzymes, which collectively constitute the purinergic landscape (Figure 2).

Three main protein families compose the so-called purinergic landscape: ectonucleotidases, P2 and P1 purinoreceptors. Ectonucleotidases are involved in eATP hydrolysis to produce adenosine (ADO). The P2 purinoreceptors are activated by eATP, and eADP. This family is subdivided into the P2X (homo-/heterotrimeric receptor channel) and P2Y (G-protein coupled) receptors. The P1 purinoreceptors are G-protein coupled receptors activated by adenosine. This family is composed of 4 members: ADORA1, ADORA2A, ADORA2B, and ADORA3.

## 3. The Purinergic Landscape in NSCLC

### 3.1. Expression of Ectonucleotidases 

Once released into the extracellular space, the fate of eATP depends on ectonucleotidases. Four enzyme families have been discovered, cloned, and characterized: ecto-nucleoside triphosphate diphosphohydrolase (NTPDases), alkaline phosphatases, ecto-nucleotide pyrophosphatase/phosphodiesterases of NPP type (NPP-type), and ecto-5′-nucleotidase [27]. In combination with the intracellular NAD-degrading enzyme CD38, enzymes for the nucleotide hydrolysis are particularly important in cancer, but also in aging [28]. CD39 (NTPD-1) and CD73 (NT5E) play an important role in calibrating the specificity, duration, and strength of purinergic signals by converting ATP/ADP to AMP, and AMP to adenosine, respectively. 

CD39, encoded by NTPDase-1, is the rate-limiting enzyme that hydrolyzes ATP to ADP and ADP to AMP [27]. CD39 is expressed by B cells, innate cells, regulatory T cells, and activated CD4 and CD8 T cells, and has been identified as a marker of exhausted T cells in patients with chronic viral infections [29,30]. In addition to immune cells, CD39 is also expressed on the tumor-associated endothelium and tumor cells [31]. The expression of CD39 was examined in 12 patients after lung surgery, and CD39 expression was found to be increased in the immune infiltrate of the tumor compared with the adjacent non-tumor tissue [32]. Specifically, TCD4+, TCD8+, FoxP3+ regulatory T cells, CD16+ NK cells, B cells, and macrophages expressed CD39 at higher levels. Remarkably, these cells also expressed higher levels of PD-1. The limited number of patients in this study did not allow for the formal conclusion that expression of CD39 and PD-1 is a marker of poor prognosis, but the three out of five patients who relapsed had a high frequency of double-positive CD39+/PD-1+ CD8+ and CD4+ TILs, suggesting that expression of CD39 in cytotoxic T cells may be an important mechanism for tumor-induced immunosuppression in NSCLC. Interestingly, quantification of the proportion of total CD8+ and CD39+ lymphocytes by immunochemistry (IHC) of patients with NSCLC was not predictive of response to ICIs. However, the double positive CD39+ CD8+ fraction appears to be a strong predictive biomarker [33]. If confirmed, this finding will be of great importance as there is an urgent need to identify patients with LC who will benefit from ICIs alone or in combination with other treatments, including anti-CD39 antibodies.

CD73, encoded by NT5E, is essential for the generation of extracellular adenosine from AMP. Adenosine could also be formed via the non-canonical pathway of CD38-CD203a-CD73, which is independent of CD39 [34]. CD73 is expressed on the surface of endothelial, stromal, and immune cells, as well as on tumor cells of various origin [35]. In patients with NSCLC, CD73 is expressed on cancer cells, cancer-associated fibroblasts (CAFs), and tumor-infiltrating lymphocytes (TILs), and its expression correlates with the expression of hypoxia- inducible factor-1, a trend confirmed in in vitro studies using the lung cancer cell lines H1299 and A549 [36]. In this study, the authors also characterized the expression of CD39 and showed that it is predominantly expressed by CAFs and TILs, in contrast to CD73, which is expressed by both cancer cells and CAFs. Overall, their results suggest that expression of CD73 and CD39 in the tumor stroma regulates immunosuppressive pathways by promoting the prevalence of FoxP3+ and PD-1+ lymphocytes as well as PD-L1 expression by cancer cells, all suggestive of an immunosuppressive microenvironment. However, no association was found between expression of ectonucleotidase and histopathological variables or survival analysis, in contrast to a previous study showing that high CD73 expression was an independent indicator of poor prognosis for overall survival and recurrence-free survival [37].

While it is widely accepted that lung cancer immune escape, tumorigenesis, and tumor progression are associated with high levels of adenosine, PD-1, and PD-L1 within the TME, it is clear that additional work is needed to fully unravel the relationship between all of these players and to fully understand whether their expression can be considered as powerful biomarkers that could guide the choice of treatments. 

Another level of complexity lies with patients whose cancer has oncogenic mutations, as highlighted in a recent study that pooled three cohorts of NSCLC patients (*n* = 4189 total) with oncogenic alterations, including KRAS, MET, RET, BRAF-V600E and non BRAF-V600E, ROS1, ALK, EGFR exon 20, HER2, and classical EGFR (exon 19 deletion and exon 21 L858R) [38]. With regard to KRAS mutations, corresponding to the largest subgroup of oncogenic lung adenocarcinoma, it has been shown that the co-occurrence of genomic alterations in the STK11 and KEAP1 genes leads to a worse outcome in KRAS-mutated patients treated with immunotherapy [39]. In addition, the impact of the tumor mutation burden (TMB) and PD-L1 expression on the clinical outcome of ICI therapies has been demonstrated for NSCLC with BRAF mutations, while EGFR and HER2 mutations and ALK, ROS1, RET and MET fusions define NSCLC subgroups with minimal benefit from ICI, despite a high level of expression of PD-L1 in NSCLC with oncogene fusions. The mechanisms underlying the lack of efficacy of ICI in EGFR-mutated NSCLC patients appear to be related to an immune-influenced phenotype characterized by a low level of expression of PD-L1, low TMB, lower cytotoxic T cell numbers, and low T cell receptor clonality. The analysis of 75 immune checkpoint genes, NTE5 (CD73), and adenosine A1 receptor (A1R) were the most upregulated genes in EGFR-mutated tumors. A single-cell analysis revealed that the tumor cell population expressed CD73, in both treatment-naïve and resistant tumors [40]. Therefore, the CD73/adenosine pathway was identified as a potential therapeutic target for EGFR-mutated NSCLC, and there is no doubt that the information from profiling the TME and antitumor immune response can be used to tailor immunotherapy in selected patients with LC.

### 3.2. Expresssion of the Purinergic Receptors

The purinergic receptor family is composed of two subfamilies; the G-protein-coupled adenosine receptors (A1R, A2RA1, A2RA2, A3R), which belong to the P1 family, and the ATP receptors, which form the P2 family. The P2 family includes the ligand-gated ion channel family receptors (P2RX1–7) and the G protein-coupled receptor (GPCR) family (P2RY1-2, P2RY4, P2RY6, and P2RY11–14) [41]. A decade ago, Geoffrey Burnstock, who was one of the first to substantiate the importance of purinergic signaling in tumor progression, hypothesized that purinergic signaling might contribute to respiratory diseases [42].

#### 3.2.1. The Adenosine Receptors

Adenosine (ADO) receptors (A1, A2A, A2B, and A3) are expressed in various cells and tissues throughout the body and are activated by ADO in the nanomolar range, with the exception of A2B, which is a low-affinity receptor (micromolar range) and therefore activated mainly under pathophysiological conditions [43]. Activation of ADO receptors on cancer cells affects proliferation, apoptosis, cytoprotection, and migration [44]. Moreover, by inhibiting the antitumor function of both lymphocytes and antigen-presenting cells, ADO promotes tumor growth [45,46]. In addition, ADO can suppress T cell priming by acting directly on DC and macrophages [47]. A2AR signaling has also been shown to block the formation of Th1 and Th17 cells and induce the development of Treg cells [48,49].

The involvement of ADO receptors in progression of LC has been poorly described. A1, A2A, A2B, and A3 receptors are expressed by type I and II alveolar epithelial cells, smooth muscle cells, endothelial cells, and immune cells [42]. Particular attention has been paid to A2AR, which is expressed by T cells in hypoxia and was initially described as a nonredundant immunosuppressive mechanism to protect normal tissues from inflammatory damage and autoimmunity. However, it quickly became clear that tumor cells exploit this immunosuppressive mechanism to their own advantage [50].

#### 3.2.2. The Purinergic Receptors

The P2Y receptor family in NSCLC. The subtypes P2RY1, P2RY2, P2RY4, P2RY6, and P2RY11–14 are expressed in mammals. These G protein-coupled receptors, that stimulate Gq- or Gi-dependent cell signaling, are activated by ligands such as extracellular ATP, ADP, UTP, UDP, UDP-glucose, and also NAD. All these ligands have different affinities for each subtype [51]. In lung cancer cells, especially in the most studied A549 cell line, the expression of P2RY-2, -6, -12 and -14 has been described. A mitogenic effect through ATP/UTP-mediated activation of P2RY2 and the UDP-activated P2RY6 was reported first [52]. Two years later, P2RY14 expression was detected in primary human type II alveolar epithelial cells, in normal bronchial epithelial cells (Beas-2b cell line), and in A549 cells. 

The P2X receptor family in NSCLC. This family consists of seven members (P2RX1–7). These ATP-gated ion channel receptors are formed by three homomeric or heteromeric P2RX subunits. Their activation directly leads to Na^+^ and Ca^2+^ influx and K^+^ efflux across the plasma membrane of the cell, which in turn activates downstream signaling pathways and triggers action potential in excitable cells (e.g., neurons), but also cell proliferation, differentiation, and apoptosis in non-excitable cells (e.g., epithelial cells) [53]. The expression of P2RX receptors mRNA in normal (Beas-2b) and tumor cell lines (H23 and A549) was analyzed by RT-qPCR. The results showed that P2RX-3 to 7 were expressed. P2RX3, 4 and 5 are overexpressed in cancer cells compared to normal cell lines, while P2RX6 and 7 are downregulated [54]. The observation that P2RX7 expression is downregulated in cancer cells is in contradiction with other studies reporting its expression. For example, TGF-β1-induced A549 migration, but not proliferation, is dependent on vesicular exocytosis of ATP, which in turn causes P2RX7 activation through actin fiber formation [55]. This result was confirmed in an independent report using 2 other lung cancer cell lines (lung adenocarcinoma PC-9 cells and mucoepidermoid carcinoma H292 cells), in which the authors also showed that P2RX7 was overexpressed in cancer cells compared with normal bronchial epithelial Beas-2B cells. Furthermore, the authors showed that in PC-9 cells, which have the highest expression of P2RX7, ATP is constitutively released and induced cell proliferation in an autocrine manner. Of note, the expression of P2RX7 in PC-9 cells is dependent on EGFR signaling, which is constitutively activated in this cell line [56]. It has been also shown that increased RNA expression of P2RX4 and P2RX7 correlated with the presence of distant metastases in NSCLC patients, although the status of EGFR mutations was not reported in this study [57]. In addition to migration, eATP has been shown to promote epithelial-to-mesenchymal transition following P2RX7 activation in A549 cells [58]. As mentioned previously, P2RX4 is also expressed by lung tumor cell lines, but its precise role in cancer progression is still unknown. 

Despite many conflicting findings in the literature, it seems clear that extracellular nucleotides are actively involved in tumorigenesis by stimulating purinergic receptors expressed directly on tumor cells or on their neighboring stromal and immune cells. The published discrepancies regarding P2RX7 expression may be related to the tools used to characterize its expression. Indeed, there are several oligonucleotides on the market, some of which recognize both the full-length and truncated mRNA of P2RX7, whereas others are specific for one or the other form. The same conclusion can be drawn for antibodies. Some of them recognize the extracellular loop present in both the full-length protein and the truncated form, while others are specific for the C-terminal domain lost in the truncated P2RX7B isoform. For example, we were unable to detect P2RX7 expression on A549 cells when using the monoclonal antibody characterized by Buell [59], whereas the use of a commercial polyclonal antibody by other authors detected its expression on various NSCLC (including A549) and SCLC cancer cell lines [22]. In the same way, tumor cells from NSCLC patients were not stained with the conformational antibody, while immune cells from the same patients showed strong staining [60]. To add complexity, some tools can also detect the truncated isoform of P2RX7 encoded by the splice variant P2RX7B, which has been shown to behave like a protumor factor [60,61].

## 4. The Purinergic Landscape: A Therapeutically Targetable Circuitry to Treat NSCLC

It is now generally accepted that ectonucleotidases and purinoreceptors, with respect to the purinergic landscape, jointly control tumor growth and metastatic spread. Indeed, exciting preclinical studies have been conducted in recent years in which players of the purinergic landscape have become novel targets in the fight against LC. For example, blocking antibodies against human CD39 and CD73 promote antitumor immunity by stimulating DC and macrophages, which in turn restore effector T cell activation [62]. In addition, an independent anti-CD39 antibody was developed and tested in various mouse tumor models. This antibody alone attenuates tumor growth, and when combined with the PD-1 antibody, it further slows tumor progression; 50% of mice show complete tumor rejection via a mechanism associated with P2RX7/NLRP3/IL-18 activation in myeloid cells [63]. As for ADO receptors, the use of PBF-509, an A2AR antagonist, in combination with ICI restores the immune response (interferon γ production) of infiltrating lymphocytes from patients with lung tumors [64]. Moreover, the interference with P2RX7 may be particularly relevant, as shown by the numerous preclinical studies described in detail by Lara and coworkers [65]. Only a few studies were dedicated to the lungs. The first showed that the P2RX7 antagonist, AZ10606120, efficiently inhibited the growth of malignant pleural mesothelioma cell lines of patients, when the tumor cell lines were implanted in immunodeficient mice [66]. This finding was attenuated by results showing accelerated tumor growth in *p2rx7*^−/−^ mice [67,68], highlighting the importance of P2RX7 expression in host cells. Consistent with this later finding, we recently published results showing that a positive allosteric modulator of P2RX7, in combination with immunotherapy, efficiently inhibited tumor growth in transplantable and oncogene-driven lung tumor models [69,70]. Moreover, purinergic receptors have been shown to play an important role in metastatic spread. Indeed, the migration of cancer cells to a secondary site in the host body is supported by activated platelets that express the P2RY12 receptor. Ticagrelor, a specific inhibitor of P2YR12 commonly used as an anti-platelet agent, inhibits metastasis of B16-F10 melanoma and 4T1 breast cancer in mouse lungs. This result was the first demonstration of the potential of pharmacological blockade of P2RY12 to prevent lung metastasis [71,72]. To our knowledge, there is no study reporting the use of ticagrelor in NSCLC patients. This may change, as there are reports showing that P2Y12-mediated nucleotide signaling plays a key role in the inflammatory response and that the administration of P2RY12 inhibitors to patients with arterial disease does not increase the risk of cancer [73,74]. In addition, there is evidence that the P2RY6 receptor acts as a negative regulator of NK cell function and inhibits their maturation and antitumor activities, which may favor the use of specific inhibitors to enhance the antitumor activity of NK cells [75].

The results of these preclinical studies have attracted great interest from pharmaceutical companies, as evidenced by the number of clinical trials registered on the clinicaltrials.gov website since 2009. Of the 63 clinical trials dedicated to LC and immunotherapies, 25 used inhibitors of proteins from the purinergic landscape (Table 1). Ten trials were dedicated to the blockade of CD39 or CD73, using drugs or monoclonal antibodies alone or in combination with ICIs (anti PD-1 or anti PD-L1), anti A2AR, or chemotherapies. All of these studies are either ongoing in phase 1, or in the first phase of recruitment, and no results have been published yet. Twelve clinical trials, using antagonists of the A2A receptor, are ongoing. Most of these trials are in the recruitment phase and aim to test the efficacy of A2A receptor antagonists in combination with other therapies. For example, the NCT03337698 study is a Phase Ib/II multicenter, randomized, large cohort study of 435 stage IV NSCLC patients, designed to evaluate the efficacy and safety of multiple immunotherapy-based treatment combinations, testing atezolizumab with 14 different molecules, including the A2AR antagonist CPI-444. This is an ongoing study that is expected to be completed by the end of 2023. Two studies have already been completed. The NCT02080078 trial combines erlotinib, a tyrosine kinase inhibitor, with theophylline, a drug with a narrow therapeutic margin that acts as an antagonist of A2AR. The main goal is to see if blocking A2AR activity can reduce the side effect of diarrhea. The study is now complete, but no results have been published yet. In the NCT02403193 trial, the authors used the A2AR antagonist NIR-178 (PBF-509) alone or in combination with anti PD-1 antibodies for NSCLC patients with advanced cancer who had failed standard therapies, including checkpoint inhibitors and TKI therapies. Promising results were observed, with good tolerability and clinical benefit of the molecule administered alone or in combination with anti PD-1/PD-L1 therapy, regardless of the PD-L1 status [76,77]. In parallel with A2AR inhibition, A3 receptor activation appears to be an alternative therapeutic option as it stimulates apoptosis in A549 human lung cancer cells [78]. Of note, stimulation of A3R by ADO also induces apoptosis in the Lu-65 giant cell carcinoma line and SBC-3 human small cell lung cancer [79,80] and CI-IB-MECA, a selective agonist of A3R, inhibits Lewis lung adenocarcinoma growth in vivo [81]. To our knowledge, A3R agonists have not yet been tested in NSCLC patients.

There are numerous small molecules and biologics targeting P2RX7, mainly developed to inhibit the receptor, that have been previously tested in humans to combat inflammatory and mood diseases [65]. We only identified 4 studies when we entered “lung cancer” and “suramin” as keywords across the 24 nonselective and selective inhibitors targeting P2RX7. Suramin is a competitive inhibitor that interacts with the ATP-binding site and potentiates the activity of chemotherapeutic agents. In 2010, suramin was tested in combination with docetaxel and gemcitabine in previously chemotherapy-treated patients with advanced NSCLC. The results of this phase 1 trial showed a manageable toxicity profile and preliminary evidence of antitumor activity [82,83]. In addition, promising results were described with an antibody that specifically recognizes the E200 epitope of non-functional P2RX7 for the local treatment of primary basal cell carcinoma lesions [84].

Undoubtedly, all these results attest to the interest in targeting players of the purinergic landscape to combat NSCLC. Nevertheless, the search for biomarkers of response will be crucial, and we believe that further combinatorial schemes need to be tested in the future. Indeed, thanks to in silico analyses, we have shown that the molecular pattern of the “purinergic genes”, namely *P2RX7*, *NT5E* (CD73), *ENTPD1* (CD39), *ADORA2A*, *ADORA2B,* and *ADORA3*, is very heterogeneous in both the 135 NSCLC cell lines and the 995 NSCLC patients available in the cBio Cancer Genomic Portal [85,86] (unpublished results). 

## 5. Conclusions

We have provided evidence for the potential of the purinergic landscape to shape antitumor responses. There are numerous molecules that modulate the activity of purinergic players, and it is likely that many of them are still under development. More importantly, some of them are already being tested in the clinic and the results of these clinical trials will undoubtedly provide new treatment options in the near future. However, given the number of players, their pattern of expression, their expression level, and their mode of action, we believe that studying their expression, distribution, and function will be helpful in determining the most efficient treatment for cancer patients, and this treatment will certainly combine multiple molecules administered together or sequentially. In this context, multiplex IHC analyses, artificial intelligence, and bioinformatics will undoubtedly be very helpful.

## Figures and Tables

**Figure 1 cancers-14-01926-f001:**
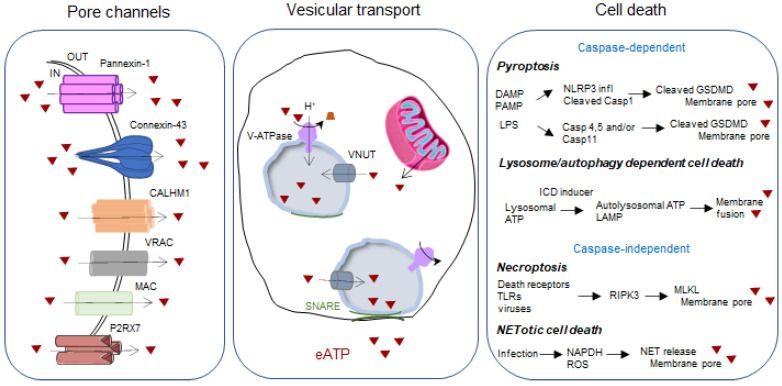
Schematic illustration of the mechanisms involved in the release of eATP.

**Figure 2 cancers-14-01926-f002:**
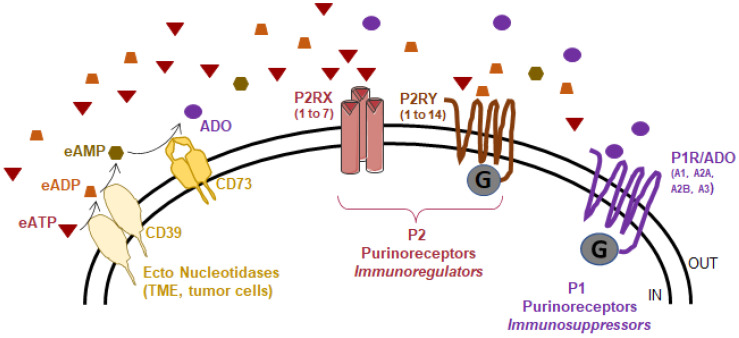
The purinergic landscape.

**Table 1 cancers-14-01926-t001:** Small molecule inhibitors and biologics used against players of the purinergic landscape in clinical trials dedicated to NSCLC.

Clinical Trial NCT	Targeted Protein	Name of Compound	Treatment	Phase	Results
04306900	CD39	TTX-30	±ICI, chemo	1	Active
04336098	CD39	SRF-617	±ICI, chemo	1	Recruiting
05143970	CD73	IPH-5301 (Ab)	±ICI, chemo	1	Recruiting
05001347	CD73	TJ004309 (Ab)	±ICI	2	Recruiting
04148937	CD73	LY3475070	±osinertinib	1	Active
03381274	CD73	MEDI-9447 (Ab)	±ICI	1	Recruiting
04672434	CD73	Syn-024 (Ab)	±ICI	1	Recruiting
03549000	CD73	NZV-930 (Ab)	±ICI	1/1b	Recruiting
03454451	CD73/A2AR	CPI-006 (Ab)	±ICI	1/1b	Recruiting
03549000	CD73/A2AR	NZV-930/NIR178	±ICI	1/1b	Recruiting
02403193	A2AR	PDF-509 (NIR178)	±ICI	1/1b	Completed with Clinical benefit [76,77]
03207867	A2AR	PDF-509 (NIR178)	±ICI	2	Recruiting
05060432	A2AR	EOS-448	±ICI	1/1b	Recruiting
04969315	A2AR	TT-10	±ICI	1/2	Not yet recruiting
03629756	A2AR	AB-298 (Etrumadenant)	±ICI	1	Completed, no results
03381274	A2AR	AZD-4635	±ICI, targeted therapy	1/2	Active
04262856	A2AR/A2BR	AB-298 (Etrumadenant)	±ICI	2	Recruiting
03846310	A2AR/A2BR	AB-298 (Etrumadenant)	±ICI	1/1b	Active
03274479	A2AR	PDF-1129		1	Recruiting
05234307	A2AR	PDF-1129	±ICI	1	Not yet recruiting
03337698	A2AR	CPI-444	±ICI, targeted therapy, chemo, radio	1b/2	Recruiting
02080078	A2AR/A2BR	Theophylline	±TKI	1	Completed, no results
01038752	P2Rs	suramin	±chemo	2	Completed
00006929	P2Rs	suramin	±chemo	2	Completed
00066768	P2Rs	suramin	±chemo	1	Completed
01671332	P2Rs	suramin	±chemo	2	Completed

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
