# Peer review of "The Purinergic Landscape of Non-Small Cell Lung Cancer"

_cancers, 2022, doi:10.3390/cancers14081926_

Round 1

Reviewer 1 Report

The authors of the review titled "The purinergic landscape in Non-Small Cell Lung Cancer" have summarized the role of purinergic landscape in non-small cell lung cancer progression and therapeutic avenues. The review is short and well-written however, with some errors such as at line 86 "activation?".

The substantial part of the review focuses on purinergic landscape and eATP, while only a small fraction focuses on therapeutic relevance of this landscape. Given that the authors have mentioned in their abstract and towards the end of their introduction, the aim of this review is to discuss how the purinergic landscape is a new avenue for new therapeutics, the amount of work in that part is not substantial. The purinergic landscape is a well reviewed topic and the novelty o this review lies in the therapeutic section of the review. So I would suggest that the authors extend their review in the side of therapeutic relevance of purinergic landscape.

Author Response

The authors of the review titled "The purinergic landscape in Non-Small Cell Lung Cancer" have summarized the role of purinergic landscape in non-small cell lung cancer progression and therapeutic avenues. The review is short and well-written however, with some errors such as at line 86 "activation?".

We thank the reviewer for his/her comment. Indeed, a problem in formatting occurred when we pasted our text into the Cancer template. The revived version addresses this problem, as well as careful review of the English language.

The substantial part of the review focuses on purinergic landscape and eATP, while only a small fraction focuses on therapeutic relevance of this landscape. Given that the authors have mentioned in their abstract and towards the end of their introduction, the aim of this review is to discuss how the purinergic landscape is a new avenue for new therapeutics, the amount of work in that part is not substantial. The purinergic landscape is a well reviewed topic and the novelty o this review lies in the therapeutic section of the review. So I would suggest that the authors extend their review in the side of therapeutic relevance of purinergic landscape.

As suggested by Reviewer 1, we have reorganized our manuscript and have expanded the section on therapeutic trials that now includes only a short survey on preclinical studies, since this subject was extensively reviewed elsewhere, and a new Table highlighting the clinical trials dedicated to the proteins of the purinergic landscape.

Reviewer 2 Report

Vouret-Craviari and colleagues propose a review to highlight the role of purinegic checkpoint in NSCLC and the outcome of immunotherapies and treatment in general.

The topic is of interest since the pathway still need to be completely unraveled and characterize. Unfortunately, the paper is not easy to read.

It appears that has been some formatting problem that make difficult to understand the point. For example, the sentences in line 72 and 85 are truncated. In Line 162 is not clear if the sentence is completed or not.

It is therefore necessary to address this problem (together with a careful review of the English language) to make the review accessible and understandable before a proper review.

The last section (lane 426 to the end) also appear to present original data that does not belong in a review article. If the data presented are not originale it should be better explained.

Author Response

Vouret-Craviari and colleagues propose a review to highlight the role of purinegic checkpoint in NSCLC and the outcome of immunotherapies and treatment in general.

 The topic is of interest since the pathway still need to be completely unraveled and characterize. Unfortunately, the paper is not easy to read.

We have reorganized the manuscript into 4 sections. The first section introduces lung cancer, the therapies currently used in patients and the need for new approaches for resistant patients. We are convinced that targeting the proteins belonging to the purinergic landscape is a promising strategy.

The second section is dedicated to the mechanisms allowing the production of extracellular ATP, eATP being at the top of the purinergic signaling chain.

In the third section, we quickly present the proteins of the purinergic landscape, with special emphasis on lung.

In the fourth section we review the evidence supporting the hypothesis that targeting the proteins of the purinergic landscape represents an interesting direction for the treatment of patients with lung cancers resistant to existing therapies and we list, in a new Table 1, the clinical trials dedicated to the proteins of the purinergic landscape. Most of these trials are in the recruitment phase, highlighting the increasing interest of the pharmaceutical companies.

It appears that has been some formatting problem that make difficult to understand the point. For example, the sentences in line 72 and 85 are truncated. In Line 162 is not clear if the sentence is completed or not. It is therefore necessary to address this problem (together with a careful review of the English language) to make the review accessible and understandable before a proper review.

Indeed, when we pasted our text in the Cancer template a formatting error occurred. The revived version addresses this problem, as well as careful review of the English language.

The last section (lane 426 to the end) also appear to present original data that does not belong in a review article. If the data presented are not originale it should be better explained.

We have removed the original figure 3.

Reviewer 3 Report

In order to bring a better version of the manuscript, authors must follow up on the recommended points and justify the evaluation.

  1. English correction and text font sizes
  2. Image resolution and image text font is not similar with the text font
  3. Full forms are missing in text
  4. There are several syntax, lexical error, grammatical error. Many of the sentences are incomplete

Author Response

we edited a new version of the manuscript, following the demand of the editor

Round 2

Reviewer 2 Report

Vouret-Craviari and colleagues present an improved version of the previous manuscript.

All the points raised have been addressed and the current version is easily readable and understandable. 

It is, therefore, suitable for publication in the present form.

Author Response

We thank you for this good news